# Clinical Outcomes of a Non-Compliant Balloon Dilatation Catheter: MOZEC™ NC Study

**DOI:** 10.3390/ijerph192316231

**Published:** 2022-12-04

**Authors:** Akshyaya Pradhan, Pravesh Vishwakarma, Monika Bhandari, Rishi Sethi, Sharad Chandra, Gaurav Chaudhary, Akhil Sharma, Marco Alfonso Perrone, Sudhanshu Dwivedi, Varun Narain

**Affiliations:** 1Department of Cardiology, King George’s Medical University, Lucknow 226003, India; 2Department of Cardiology and Cardio Lab, University of Rome Tor Vergata, 00133 Rome, Italy

**Keywords:** balloon angioplasty, coronary stenosis, non-compliant balloons, percutaneous coronary intervention

## Abstract

The present study sought to assess the clinical outcomes of the Mozec™ Non-compliant (NC) Rx PTCA balloon dilatation catheter (BDC) (Meril Life Sciences Pvt. Ltd., Vapi, India) for dilatation of coronary lesions. This was a post-marketing, single-centre, single-arm, retrospective study. In total, 57 patients who had undergone post-dilatation with the Mozec™ NC Rx PTCA balloon dilatation catheter were evaluated. The primary endpoint was procedural success defined as (i) successful delivery of the investigational device to and across the target lesion; (ii) successful inflation, deflation, and withdrawal of the investigational device; (iii) absence of vessel perforation, flow-limiting vessel dissection, increase in thrombolysis in myocardial infarction (TIMI) flow from baseline, clinically significant arrhythmia requiring medical treatment; and (iv) achievement of final TIMI flow grade 3 after percutaneous coronary intervention of the target lesion after single or multiple attempts to cross the target lesion. Procedural success was achieved in 57 (100%) patients. There were no incidences of major adverse cardiac events (MACE)/target lesion failure (TLF). Mozec™ NC Rx PTCA balloon dilatation catheter has demonstrated favourable outcomes for the dilatation of routine and complex coronary lesions in a small cohort, as evidenced by its 100% procedural success rate and absence of MACE.

## 1. Introduction

Percutaneous transluminal coronary angioplasty (PTCA)—the legacy of Andreas Grüntzig—began more than four decades ago. Balloon catheters were the first devices pioneered to re-open occluded atherosclerotic coronary arteries and have since paved the way for percutaneous coronary interventions (PCI) with the implantation of drug-eluting stents [1]. Nonetheless, balloon catheters remain the primary workhorse device in such interventions and are still adequately employed for both pre- and post-dilatation of atherosclerotic lesions [2].

Over the years, compliant balloon catheters have gradually evolved into semi-compliant (SC), non-compliant (NC), super high-pressure, cutting/scoring and micro or miniature balloons whilst, catheters have been designed to be more steerable and allow system exchange. The compliance of a balloon catheter primarily depends on the increased diameter that results from a predetermined increase in inflation pressure [3]. SC balloons respond to an increase in inflation pressure and are only partially able to conquer the significant lesion resistance [4,5]. In contrast, NC balloons expand uniformly and typically cannot be expanded beyond a predetermined diameter limit [6]. Thus, NC balloons provide an additional benefit of deforming in a more controlled and stable manner during inflation [5,6,7].

In recent years, these NC balloons have been used for pre-dilatation in cases of rigid calcified lesions prior to stenting and post-dilatation of the implanted stent for optimization of stent parameters, bifurcation treatment by kissing balloon technique [8] or treatment of restenosis [9]. In comparison to SC balloons, NC balloons exhibit more consistent outcomes. Further, when dilating a resistant lesion, applying pressure over the permissible threshold exacerbates the probability of non-uniform balloon expansion, leading to over-dilatation of the lesion margins (termed as “dog-boning”) and thereby increases the risk of vascular damage and likelihood of restenosis [6,10]. Nevertheless, despite attaining optimal stent expansion and a significant outcome, several post-dilation studies have been unable to show distinct long-term advantages of both types of balloons. Considering these all aspects, research into the innovative balloon dilatation catheter is worthwhile in order to obtain the best stent expansion and the largest possible luminal area particularly in complex lesion subset.

Mozec™ NC Rx PTCA balloon dilatation catheter (BDC) (Meril Life Sciences Pvt. Ltd., Vapi, India) is a sterile single rapid exchange catheter specifically designed for post-dilatation. It consists of a folded NC balloon, a soft tip, a distal shaft of two lumens, and a proximal shaft with single lumen; used for dilatation of the stenotic portion of a coronary artery or a bypass graft stenosis in order to improve myocardial perfusion and post-delivery expansion of balloon-expandable stents. Mozec™ NC Rx PTCA BDC has been approved by the US Food and Drug Administration (FDA) in May 2017. The objective of the present study was to assess the clinical outcomes of the Mozec™ NC Rx PTCA BDC for dilatation of coronary lesions.

## 2. Materials and Methods

### 2.1. Study Design and Patient Population

This was a post-marketing, single-centre, single-arm, retrospective study carried out at the Department of Cardiology, King George’s Medical University, Lucknow, Uttar Pradesh, India. A total of 57 patients above the age of 18 years who required dilatation of stenosis in coronary artery or a bypass graft using the Mozec™ NC Rx PTCA BDC were enrolled in the study. The study protocol was approved by the Institutional Ethics Committee of King George’s Medical University (ECR/262/Inst/UP/2013/RR-16). The study design and procedures conformed to the principles of the Declaration of Helsinki [11] and the ICH-GCP guidelines [12]. All study patients provided written informed consent to undergo the procedure and for their data to be used for clinical research.

### 2.2. Study Device

The Mozec™ NC Rx PTCA BDC is a non-compliant, rapid exchange PTCA BDC that is specifically designed to dilate stenotic lesions of a coronary artery or graft and to expand balloon-expandable stents post implantation. Its nylon construction and the soft tip of the catheter facilitates access of the catheter through the stenotic lesion. In order to place the balloon precisely under fluoroscopy, the radiopaque marker band of the balloon indicates the approximate balloon operating length at nominal pressure. The coaxial lumens permit both movement of the guidewire (5F) as well as balloon inflation. Two markers on the proximal shaft indicate the catheter position in relation to the brachial (90 cm) or femoral (100 cm) guiding catheter tip. The proximal portion of the shaft is made up of a polytetrafluoroethylene coating. The 0.019″ distal tip entry facilitates access into the constrictive lesions. This balloon is available in lengths ranging 8–38 mm and diameters ranging 2.00–5.00 mm. The Mozec™ NC Rx PTCA balloon dilatation catheter is illustrated in Figure 1.

### 2.3. Interventional Procedure and Concomitant Medications

According to standard ACC/AHA guidelines, PTCA procedure was performed [13,14]. All procedures were performed via the 6F guiding catheter and the radial access or femoral access point was preferred, as required. Pre-operatively, post-operatively, and on follow-up, all patients were received a standard antiplatelet regimen (aspirin 150 mg daily and clopidogrel 75 mg daily). The use of GP IIb/IIIa inhibitors was left to the operator’s discretion. All serious adverse events (SAE) were reported to the regulatory bodies as per guidelines.

### 2.4. Study Endpoints and Study Definitions

The primary endpoint was procedural success, defined as (i) successful delivery of the investigational device to and across the target lesion; (ii) successful inflation, deflation, and withdrawal of the investigational device; (iii) absence of vessel perforation, flow-limiting vessel dissection, improvement of thrombolysis in myocardial infarction (TIMI) flow from baseline, or clinically significant arrhythmia requiring medical treatment; and (iv) achievement of final TIMI flow grade 3 after PCI of the target lesion following single or multiple attempts to cross the target lesion [15]. Secondary endpoints were: (i) in-hospital major adverse cardiac events (MACE); (ii) target lesion failure (TLF); (iii) cardiac death and non-cardiac death; (iv) stent thrombosis (ST); (v) target vessel revascularization (TVR) not derived from ST; (vi) significant vessel dissection; (vii) device-related death; (viii) device-related death and myocardial infarction (MI); (ix) total procedure time; (x) number of attempts to cross the lesion, and inflation/deflation cycles for the balloon. In-hospital MACE was defined as a composite of TVR, cardiac death, and MI (Q-wave and non-Q-wave). In-hospital TLF was defined as a composite of target lesion revascularization.

### 2.5. Data Collection

All demographic, lesion, procedural, and outcome data were collected from hospital records and retrospectively extrapolated for the current analysis. Clinical events occurring during the procedure, post-procedure, during hospitalization, and at follow-up were documented.

### 2.6. Statistical Analysis

Categorical variables are presented as numbers and percentages and were compared using Pearson’s chi-square test or Fisher’s exact test. Continuous variables are represented as mean ± SD and compared using t-test or Mann–Whitney test, as appropriate. A two-sided *p*-value < 0.05 was considered statistically significant. All data were analysed using the Statistical Package for Social Sciences (SPSS 26.0, Inc., IBM, Chicago, IL, USA).

## 3. Results

### 3.1. Baseline and Demographic Characteristics

The mean age of the study population was 55.1 ± 11.5 years. Males contributed 44 (77.2%) patients of the study population. Comorbidities such as hypertension and diabetes mellitus were present in 18 (31.6%) and 13 (22.8%) patients, respectively. Moreover, seven (12.3%) patients had a history of MI, and one (1.8%) patient had previously undergone PCI, respectively. The baseline and demographic characteristics of the study population are outlined in Table 1.

### 3.2. Lesion and Procedural Characteristics

The Mozec™ NC Rx PTCA balloon dilatation catheter was used to treat 57 of 140 lesions. The characteristics of the lesions comprised bifurcation (33%), LMCA (9%), calcified lesions (18%), tortuous lesion (12%) and Chronic Total Occlusion (CTO, 7%). A total of 10 (17.5%) lesions were total occlusions, the rest 6 being acute occlusions. Left anterior descending artery was most-commonly treated (39%) artery followed by right coronary artery (32%).

In most cases, patients presented with lesions of moderate complexity (type B1–45.7%, type B2–30.7%). Pre- and post-dilatation were performed in 57 (100%) lesions. Mean balloon length and diameter were 12.5 ± 2.4 mm and 2.8 ± 0.5 mm, respectively. Lesions were classified according to American College of Cardiology (ACC)/American Heart Association (AHA) lesion morphology criteria [15]. The lesion and procedural characteristics are detailed in Table 2 and Figure 2. The details of the stent employed, pre-dilatation and post-dilatation balloon lengths and sizes are depicted in Table 3.

### 3.3. Outcomes

Procedural success was achieved in 100% of patients. The flexibility, pushability, trackability, crossability, and deliverability of the balloon were all determined to be satisfactory, therefore the investigators did not encounter issues during inflation, deflation, or withdrawal.

There were no cases of MACE and TLF post procedure. Post-procedure TIMI flow grade 3 was achieved in 98.3% patients as illustrated in Figure 3. No significant serious adverse events have been reported until 10.40 ± 0.26 months follow-up period.

## 4. Discussion

This was a retrospective, single-centre study, in which the primary endpoint i.e., procedural success—was achieved in all the patients, in whom PTCA was performed using the study device (57 of 140 patients). Regarding the safety and efficacy of Mozec™ NC Rx PTCA BDC, the authors believe that these early results are significant, considering the high degree of vessel complexity observed, including bifurcation, calcified, and tortuous lesions and CTOs. The procedural success attained in terms of re-establishment of myocardial perfusion to TIMI flow grade 3 in a maximum proportion of patients, absence of MACE events, and the absence of post-procedural TLF are the most key findings of this study. Furthermore, neither MACE nor TLF had occurred through the 10.40 ± 0.26 months follow-up period, which establish a basis for the early clinical safety of this device.

Several studies support that post-dilatation with NC balloons following stent deployment can enhance stent expansion and perhaps reduce the incidences of TVR and ST [16]. The Pre-NC and OPRENBIS proof-of-concept studies [2] included a pooled analysis of optical coherence tomography data that revealed pre-dilatation stent expansion indexes (SIEs) of 0.88 ± 0.13 and 0.85 ± 0.14 in the NC and the SC balloon groups (*p* = 0.018), respectively. After post-dilatation, these SEIs improved to 0.94 ± 0.13 in the NC balloon group and 0.88 ± 0.13 in the SC balloon group (*p* = 0.02) [2]. Furthermore, a comparative study by Özel et al. examined the peri-procedural efficacy and long-term outcomes of NC and compliant balloons employed for pre-dilatation prior to stent implantation. The study revealed that pre-dilatation with the NC balloon not only eliminated the requirement for post-dilatation, but it decreased the time needed for procedures, fluoroscopy, and contrast volume as well [17].

The STent OPtimization (STOP) study [18] reported that post-dilatation with an NC balloon remarkably improved the stent optimization from 21% to 54% following implantation under intravascular ultrasound guidance. The POSTDIL-STEM trial also found that post-dilatation during primary PCI for ST-elevation MI enhanced the stent expansion, apposition, and post-PCI fractional flow reserve without having a significant impact on the coronary microcirculation [19]. Furthermore, post-dilatation with an NC balloon is necessary to achieve optimal expansion while effectively minimizing malposition [20].

Overall, the data from the aforementioned investigations show positive outcomes when employing NC balloon dilatation catheters and fortify the present study’s findings. In a recent first-in-man study, the safety and efficacy of the novel Balton NC balloon catheter (Balton, Poland) was examined, which reported a procedural success rate of 100% [7]. These findings are in good agreement with the outcomes of current study demonstrating 100% procedural success using the Mozec™ Rx NC PTCA BDC. It was noteworthy that the patient population of that study had 47.6% lesions in the left circumflex artery, 19.1% lesions were of type B2/C category, and multi-vessel disease was present in 61.9% patients. In this study, the majority of patients had moderately complicated lesions (type B1–45.7%, type B2–30.7%), while high-risk lesions were less common (type C–15.7%).

A high frequency of cases of coronary artery disease are encountered in the real-world scenario, in which and PCI is considered suitable for undergoing PCI and may present with acute coronary syndrome that necessitates immediate intervention in the critical coronaries. In such a scenario, these data ascertain the advantages of using the Mozec™ NC Rx PTCA BDC for a vast and vulnerable patient population. Moreover, such encouraging data offer interventional cardiologists a suitable treatment device that has the potential to prevent stent under-expansion, and consequentially in-stent restenosis and stent thrombosis.

In a pertinent case series on chronic coronary syndrome patients, Włodarczak et al. detected several undilatable lesions in the left circumflex and left anterior descending arteries [21]. The authors noted that the lesions encountered in their patients were severely undilatable for routine PCI, which necessitated rotational atherectomy in combination with lithotripsy before the post-dilatation was performed with high-pressure inflation using the 20 atm NC balloon Emerge (Boston Scientific, Marlborough, MA, USA). The findings of this case-series highlight that the degree and severity of atherosclerotic plaque are crucial factors for the success of post-dilatation using NC balloons.

The number of patients undergoing PCI has increased massively, especially as the prevalence of CAD rises in developing countries including India [22]. Such an increasing prevalence raises the health economic burden associated with CAD. In the current scenario, more cost-effective devices would aid to control the individual patient health care costs in countries such as India where healthcare needs are majorly met by out-of-pocket expenditure [23]. The Mozec™ NC Rx PTCA BDC is one such cost-effective device that can reduce the economic burden on patients undergoing PCI. National health insurance schemes have even included this device. This inclusion is supported by the affordable cost and equivalent efficacy in comparison to other more expensive competitor devices on the market. Table 4 compares the features of contemporary NC balloons that are widely used in the routine clinical settings [24,25,26,27,28,29].

### Study Limitations

There are a few limitations of the present study that should be mentioned. The retrospective, single-arm study design without a control group is the first limitation of the present study. A study comprising a larger sample size would be better able to provide insightful clinical information. The second limitation was the short follow-up duration. Thirdly, this study comprised a small cohort of complex lesion patient population. Moreover, periprocedural MI and number of attempts were not analysed and were other shortcomings of this study. Lastly, the underutilization of intracoronary imaging guidance is yet another study limitation.

## 5. Conclusions

The Mozec™ NC Rx PTCA BDC demonstrated favourable outcomes for the dilatation of both simple as well as complex coronary lesions. The achievement of final TIMI flow grade 3 following PCI and an acceptable device success rate demonstrates the efficacy of Mozec™ NC Rx PTCA BDC. The follow-up data further confirm the safety of balloon catheter in this small cohort study, since there were no incidences of MACE and TLF.

## Figures and Tables

**Figure 1 ijerph-19-16231-f001:**
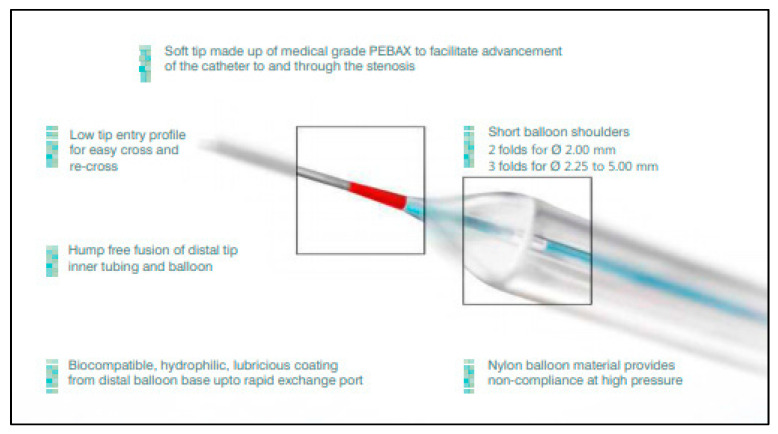
Mozec™ NC Rx PTCA balloon dilatation catheter.

**Figure 2 ijerph-19-16231-f002:**
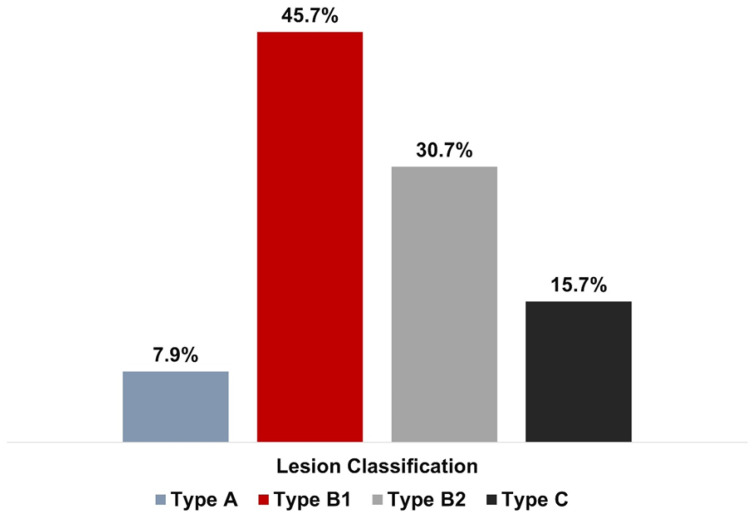
Lesion classification according to the modified American College of Cardiology/American Heart Association classification.

**Figure 3 ijerph-19-16231-f003:**
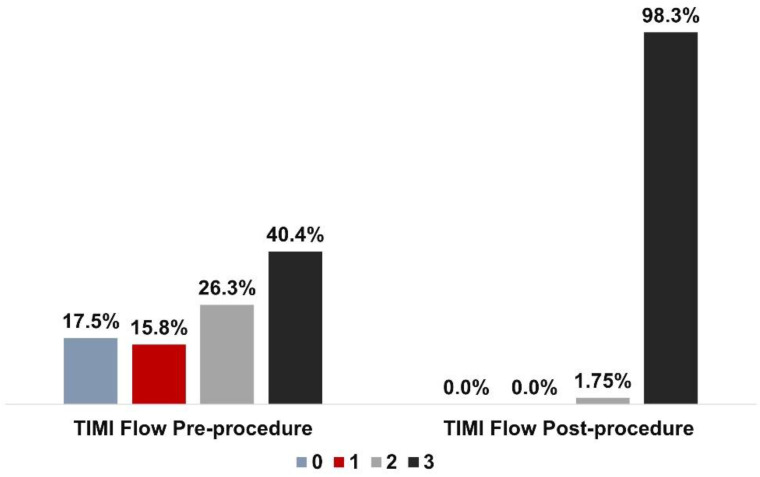
Thrombolysis in myocardial infarction (TIMI) flow pre- and post-procedure.

**Table 1 ijerph-19-16231-t001:** Baseline and demographic characteristics.

Baseline Characteristics	Patients *n* = 57
Age, years	55.1 ± 11.5
Males	44 (77.2%)
Heart rate, beats per min	77.9 ± 14.9
Systolic blood pressure, mmHg	121.1 ± 18.0
Diastolic blood pressure, mmHg	73.3 ± 17.0
Medical History	
Hypertension	18 (31.6%)
Diabetes mellitus	13 (22.8%)
Smoking	3 (5.26%)
Dyslipidaemia	2 (3.5%)
Diagnosis at Presentation Stable angina	20 (35.1%)
Acute coronary syndrome	37 (64.9%)
ST-elevation MI	17 (46)
Non-ST-elevation MI	15 (40.5)
Unstable angina	5 (13.5)
Single vessel disease	11 (20)
Dual vessel disease	23 (40)
Multivessel disease	23 (40)
Previous coronary artery bypass graft	0 (0%)
Previous myocardial infarction	7 (12.3%)
Previous percutaneous coronary intervention	1 (1.8%)

All data are expressed as number (percentage) or mean ± standard deviation.

**Table 2 ijerph-19-16231-t002:** Lesion characteristics.

Clinical Characteristics	Patients *n* = 57
Lesion details	
Total number of lesions	140
Total number of lesions treated	57
Number of lesions per patient	2.46
Percentage diameter stenosis	86.0 ± 15.0
Pre-dilatation	57 (100)
Post-dilatation	57 (100)
Lesion features	
Bifurcation	19 (33.33)
Calcified	10 (17.5)
Tortuosity	7 (12.3)
Chronic total occlusion	4 (7)
Lesion locations	
Left main coronary artery	5 (9)
Left anterior descending artery	22 (38.6)
Left circumflex artery	10 (17.5)
Right coronary artery	18 (31.6)
Ramus intermedius	06 (10.5)
Procedural details	
Procedural success	57 (100)
Post Mozec™ NC Rx balloon TIMI III flow	56 (98.3)
Major adverse cardiac event	0 (0.0%)
Target lesion failure	0 (0.0%)

All data are expressed as number (percentage) or mean ± standard deviation.

**Table 3 ijerph-19-16231-t003:** Stent, pre-dilatation and post-dilatation balloon specifications.

Parameter	*n* (%)
Total stents	86 (100)
Stent per patient	1.5
DES, *n* (%)	85 (98.8)
Everolimus-eluting stent	47 (56)
Sirolimus-eluting stent	28 (33)
Zotarolimus-eluting stent	9 (11)
BMS, *n* (%)	1 (1.2)
Balloon details	
Balloon length, mm	12.5 ± 2.4
Balloon diameter, mm	2.8 ± 0.5
Post-dilatation balloon	
Length, mm, *n* (%)	
8	5 (8.77)
10	21 (36.84)
13	16 (28.07)
15	15 (26.32)
Diameter, mm, *n* (%)	
2.50	1 (1.75)
2.75	9 (15.79)
3.00	24 (42.11)
3.50	21(36.84)
4.0	2 (3.51)

All data are expressed as number (percentage) or mean ± standard deviation.

**Table 4 ijerph-19-16231-t004:** Technical comparison of contemporary non-compliant balloon dilatation catheters.

Features	Pantera LEO (Biotronik) [24]	NC Trek Rx (Abbott) [25]	NC Emerge™ (Boston Scientific Corporation) [26]	River (Balton) [27]	Mini Trek (Abbott) [28]	NC Sprinter Rx (Medtronic) [29]	Mozec NC (Meril Life Sciences)
Nominal pressure (atm)	14	12	12	8	8	10	12
Rated burst pressure (atm)	18 & 20	18	18/20	18	14	18	20
Guidewire compatibility (inch)	0.014	0.014	0.014	0.014	0.014	0.016	0.014
Shaft length (cm)	145	143	142	140	145	138	142
Sheath compatibility (F)	5	-	5,6	5	5,6	-	5
Balloon length (mm)	8–30	6–25	6–30	10–40	6–20	6–27	8–38
Balloon diameter (mm)	2.00–5.00	1.50–5.00	2.00–6.00	1.25–4.00	1.20–2.00	2.00–5.00	2–5

## Data Availability

The data presented in this study are available on request from the corresponding author.

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
