# Peer review of "Clinical Outcomes of a Non-Compliant Balloon Dilatation Catheter: MOZEC™ NC Study"

_ijerph, 2022, doi:10.3390/ijerph192316231_

Round 1
Reviewer 1 Report
In this manuscript Pradhan et al. retrospectively analysed efficacy and safety of a novel non-compliant balloon dilatation catheter, used on 57 patient underwent percutanoeus coronary angioplasty.
The manuscript is properly written and the language is clear. Discussion and methods should be improved.
Important revisions should be made, without which the manuscript should not be taken into account for publication:
- A control group with other approved NC-balloon dilatation catether, routinely used in clinical practice, is necessary to give power to primary and secondary endpoint.
- It's useful to add other features about treated lesion: e.g. how much lesion was treated with stent, how much lesion was bifurcations.
- Endpoints reported in study definition, in particular a lot of secondary endpoint, are not described in the results (e.g. total procedure time, number of attempts): can you explain this? It could be important to add myocardial injury/MI after the procedure, inserting high-sensitivity troponin values.
- Details about statistical analysis must to be added in the text, not only in the legend of tables/figures.
Reviewer 2 Report
The authors performed a single-centre, single-arm, retrospective analysis to assess the clinical outcomes of the MozecTMPTCA balloon dilatation catheter for dilatation of coronary lesions. A total of 57 patients were enrolled in the study. Procedural success was achieved in 100% of patients. No incidence of major adverse cardiac events was reported at 10.4 ± 0.26 months of follow-up.
I have the following major concerns:
- Please specify in the baseline and clinical characteristics: the number of patients presenting with acute coronary syndrome, multivessel disease, and lesions location. Also, 17.5% of lesions were total occlusion: were they patients with chronic occlusions or ACS?
- I guess all 57 lesions were stented, but it would be better to specify it in the text. Also, it would be interesting to know which stents were used.
- Lesions details: only 57 out of 140 lesions were treated. How did you manage the remaining 83 lesions?
- Limitation section: you must specify that this was not a complex population, as reflected by low rates of previous CABG and PCI in a relatively young population (55 y.o.).
- Finally, I would scale back the conclusions (also in the abstract) by specifying that in this preliminary evaluation such procedural results were observed in a small, noncomplex population.
Minor concerns:
- Abstract: “A total of 57 patients who had underwent post-dilatation with…”. Such a sentence may cause confusion in the reader, as the balloons were also used for stenoses pre-dilation. Please turn the sentence in “A total of 57 patients who had underwent pre- and/or post-dilatation with…”.
Round 2
Reviewer 2 Report
I thank the authors for responding in detail to my comments. I found the manuscript significantly improved compared to the previous version. However, I believe there are still some points to be clarified:
- I would suggest the authors to order paragraphs in the discussion section in the following way: 1) report the main results of your study (there are few sentences that I had trouble following at the beginning of the discussion, I think detailed copyediting would be helpful), then 2) report the results of other studies on NC balloons and those of the trials comparing SC and NC balloons and finally 3) compare the results of all these studies with yours (are the results of your study in line with those of previous studies? What does your study add?).
- I would delete the second paragraph in the discussion section (from line 181 to 187): the topic has already been addressed in the Introduction.
- I am not sure I understand what is reported between lines 227 and 235 when the results of the case series by WÅ‚odarczak et al are described. Please rephrase. In general, the paper lacks of readability. I would suggest the authors to have their work edited by a native English speaker.
Minor:
- References in Table 4 are incorrect. NC Emerge is not only an over the wire PTCA catheter.
Author Response
Response to Reviewers:
Article No: ijerph1855507
Study Title: Clinical Outcomes of a Non-Compliant Balloon Dilatation Catheter: MOZEC™ NC Study
I thank the authors for responding in detail to my comments. I found the manuscript significantly improved compared to the previous version. However, I believe there are still some points to be clarified.
Reply: We sincerely appreciate your thoughtful suggestions and advice for improvements. We have indeed benefitted from your reviews and suggestions.
- I would suggest the authors to order paragraphs in the discussion section in the following way: 1) report the main results of your study (there are few sentences that I had trouble following at the beginning of the discussion, I think detailed copyediting would be helpful), then 2) report the results of other studies on NC balloons and those of the trials comparing SC and NC balloons and finally 3) compare the results of all these studies with yours (are the results of your study in line with those of previous studies? What does your study add?).
Reply: These comments have been critically helpful in re-editing the Discussion section. We have, in line with your suggestions, presented the main results first:
This was a retrospective, single-center study, in which the primary endpoint —procedural success —was achieved in all the patients, in whom PTCA was performed using the study device (57 of 140 patients). Regarding the safety and efficacy of Mozec™ NC Rx PTCA BDC, the authors believe that these early results are significant, considering the high degree of vessel complexity observed, including bifurcation, calcified, and tortuous lesions and CTOs. The procedural success attained in terms of re-establishment of myocardial perfusion to TIMI flow grade 3 in a maximum proportion of patients, absence of MACE events, and the absence of post-procedural TLF are the most key findings of this study. Furthermore, neither MACE nor TLF had occurred through the 10.40 ± 0.26 months follow-up period, which establish a basis for the early clinical safety of this device.
Thereafter, we have presented the results of other studies on NC balloons including trials comparing NC and semi-compliant (SC) balloons.
Several studies support that post-dilatation with NC balloons following stent deployment can enhance stent expansion and perhaps reduce the incidences of TVR and ST [16]. The Pre-NC and OPRENBIS proof-of-concept studies [2] included a pooled analysis of optical coherence tomography data that revealed stent expansion indexes (SIEs) of 0.88 ± 0.13 and 0.85 ± 0.14 in the NC and the SC balloon groups (p=0.018), respectively. After post-dilatation, these SEIs improved to 0.94 ± 0.13 in the NC balloon group and 0.88 ± 0.13 in the SC balloon group (p=0.02) [2]. Further, a comparative study by Özel et al examined the peri-procedural efficacy and long-term outcomes of NC and compliant balloons employed for pre-dilatation prior to stent implantation. The study revealed that pre-dilatation with the NC balloon not only eliminated the requirement for post-dilatation but it decreased the time needed for procedures, fluoroscopy, and contrast volume as well [17].
The STent OPtimization (STOP) study [18] reported that post-dilatation with an NC balloon remarkably improved the stent optimization from 21% to 54% following implantation under intravascular ultrasound guidance. The POSTDIL-STEM trial also found that post-dilatation during primary PCI for ST- elevation MI enhanced the stent expansion, apposition, and post PCI fractional flow reserve without having a significant impact on the coronary microcirculation [19]. Further, post-dilatation with an NC balloon is necessary to achieve optimal expansion while effectively minimizing malapposition [19].
Then, we compared the results of other studies and presented the significance of our findings and their impact on the literature. As we have stated in our Discussion section, several studies have elucidated the advantages of post-dilatation including the reduction in restenosis and stent thrombosis. The earlier randomized controlled trials comparing the post-dilatations by semi-compliant and non-compliant balloons showed significantly higher improvement in the stent expansion in cases when NC balloons have been used for post-dilatation in comparison to the use of SC balloons for post-dilatation, as reported by Cuculi et al who analyzed the results of two randomized studies (the Pre-NC and OPRENBIS studies). Secondly, we noted some striking insights by WÅ‚odarczak et al who concluded that the extent of lesion calcification and atherosclerotic plaque are crucial clinical factors that guide the procedural success of high-pressure post-dilatation by NC balloons. Therefore, we have restructured and corrected the Discussion section in accordance with your fruitful suggestions.
Overall, the data from the aforementioned investigations show positive outcomes when employing NC balloon dilatation catheters and fortify the present study findings. In a recent first-in-man study, the safety and efficacy of the novel Balton NC balloon catheter (Balton, Poland) was examined, which reported a procedural success rate of 100%. [7]. These findings are in good agreement with the outcomes of current study demonstrating 100% procedural success using the Mozec™ Rx NC PTCA BDC. It was noteworthy that the patient population of that study had 47.6% lesions in the left circumflex artery, 19.1% lesions were of type B2/C category, and multi-vessel disease was present in 61.9% patients. In this study, the majority of patients had moderately complicated lesions (type B1–45.7%, type B2–30.7%), while high-risk lesions were less common (type C–15.7%).
A high frequency of cases of coronary artery disease are encountered in the real-world scenario, in which and PCI is considered suitable for undergoing PCI and may present with acute coronary syndrome that necessitates immediate intervention in the critical coronaries. In such a scenario, these data ascertain the advantages of using the Mozec™ NC Rx PTCA BDC for a vast and vulnerable patient population. Moreover, such encouraging data offer interventional cardiologists a suitable treatment device that has the potential to prevent stent under-expansion, and consequentially in-stent restenosis and stent thrombosis.
In a pertinent case series on chronic coronary syndrome patients, WÅ‚odarczak et al detected several undilatable lesions in the left circumflex and left anterior descending arteries [20]. The authors noted that the lesions encountered in their patients were severely undilatable for routine PCI, which necessitated rotational atherectomy in combination with lithotripsy before the post-dilatation was performed with high-pressure inflation using the 20 atm NC balloon Emerge (Boston Scientific, Marlborough, MA, USA). The findings of this case-series highlights that the degree and severity of atherosclerotic plaque are crucial factors for the success of post-dilatation using NC balloons.
The number of patients undergoing PCI has increased massively especially as the prevalence of CAD rises in developing countries including India. Such an increasing prevalence raises the health economic burden associated with CAD. In the current scenario, more cost-effective devices would aid to control the individual patient health care costs in countries such as India where healthcare needs are majorly met by out-of-pocket expenditure [21]. The Mozec™ NC Rx PTCA BDC is one such cost-effective device which can reduce the economic burden on patients undergoing PCI. The National health insurance schemes have even included this device. This inclusion is supported by the affordable cost and equivalent efficacy in comparison to other more expensive competitor devices on the market. Table 4 compares the features of contemporary NC balloons that are widely used in the routine clinical settings [22-27].
- I would delete the second paragraph in the discussion section (from line 181 to 187): the topic has already been addressed in the introduction.
Reply: This advice was extremely useful, and we thank the reviewers for pointing this out. We have deleted this text in the revised manuscript.
- I am not sure I understand what is reported between lines 227 and 235 when the results of the case series by Wlodarczak et al are described. Please rephrase. In general, the paper lacks of readability. I would suggest the authors to have their work edited by a native English speaker
Reply: We express our gratitude for this advice. We agree that there was a lot of ambiguity in this section. Therefore, we have revised this paragraph to reflect our meaning with clarity and accuracy.
In a pertinent case series on chronic coronary syndrome patients, WÅ‚odarczak et al detected several undilatable lesions in the left circumflex and left anterior descending arteries [20]. The authors noted that the lesions encountered in their patients were severely undilatable for routine PCI, which necessitated rotational atherectomy in combination with lithotripsy before the post-dilatation was performed with high-pressure inflation using the 20 atm NC balloon Emerge (Boston Scientific, Marlborough, MA, USA). The findings of this case-series highlight that the degree and severity of atherosclerotic plaque are crucial factors for the success of post-dilatation using NC balloons.
Minor comments
- References in Table 4 are incorrect. NC Emerge is not only an over the wire PTCA catheter
Reply: We appreciate you bringing this to our notice. In this version, references in Table 4 have been updated and verified. We agree that NC Emerge is not only an over the wire PTCA catheter. This content was unintentionally included in the table; the appropriate adjustments have been made in the revised version.